# OpenReview forum: "AdvPrompter: Fast Adaptive Adversarial Prompting for LLMs"
_ICLR.cc/2025/Conference — Submitted to ICLR 2025_

### Official Review · Reviewer_qn1H · 2024-10-21

**Soundness:** 3
**Presentation:** 3
**Contribution:** 3
**Rating:** 6
**Confidence:** 4

**Summary:**

The paper titled "AdvPrompter: Fast Adaptive Adversarial Prompting for LLMs" presents a novel approach to generating adversarial prompts that jailbreak large language models (LLMs), enabling the generation of harmful or inappropriate content. Traditional methods for adversarial prompt generation, such as manual red-teaming or optimization-based methods, can be slow, inefficient, and prone to generating semantically meaningless attacks. In contrast, the authors propose AdvPrompter, an LLM trained using a novel algorithm to rapidly generate human-readable adversarial prompts without requiring gradient information from the target LLM.

The core innovation of the paper lies in its alternating training method, AdvPrompterTrain, which alternates between generating adversarial suffixes and fine-tuning the AdvPrompter model. The resulting adversarial prompts are highly effective, achieving state-of-the-art results on the AdvBench and HarmBench datasets, with improved attack success rates, faster generation times, and strong transferability to black-box LLMs. Additionally, the paper demonstrates that by fine-tuning LLMs on datasets generated by AdvPrompter, models can become more robust against jailbreaking attacks while maintaining high performance on benchmarks like MMLU and MT-bench.

**Strengths:**

1. **Clarity**: The paper is well-structured and provides clear explanations of its methodology, backed by comprehensive experimental results and ablation studies. The use of figures and tables, such as Table 1, aids in understanding the comparative advantages of the method. The training algorithm and attack framework are concisely explained, which improves accessibility.

2. **Significance**: AdvPrompter presents significant contributions to the area of LLM robustness and safety, offering a highly scalable solution to automatic red-teaming. Its gradient-free approach makes it applicable in both white-box and black-box settings, which broadens its impact for securing deployed LLM systems. The paper’s findings on fine-tuning models with adversarial data to improve safety also open up new avenues for automated adversarial training in LLMs.

**Weaknesses:**

1. **Lack of Comparison with BEAST**: One notable omission is the lack of detailed comparison with BEAST (introduced in "Fast Adversarial Attacks on Language Models in One GPU Minute")​. BEAST also focuses on gradient-free attacks and is highly efficient, achieving impressive success rates within one GPU minute, making it an essential baseline. The authors of AdvPrompter reference BEAST, but they fail to provide head-to-head benchmarks, especially in terms of speed and success rates. This limits the ability to assess whether AdvPrompter's claim of "fast" generation holds up against a method already proven to be both rapid and effective.

2. "Unclear Computational Efficiency": While AdvPrompter claims faster generation of adversarial prompts compared to gradient-based methods, the paper does not include detailed benchmarks or profiling to demonstrate computational efficiency on a per-prompt basis. For instance, BEAST reports precise GPU utilization metrics and compares the attack time per prompt across different models. AdvPrompter lacks such concrete data, making its claims of speed improvements less convincing. Without these comparisons, it is unclear whether the method is truly fast or simply optimized for a limited set of tasks.

Incomplete Discussion of Attack Readability: While AdvPrompter claims to generate human-readable adversarial prompts, there is limited qualitative analysis of the readability or coherence of these prompts. I would like to see more analysis on this.

**Questions:**

I encourage the authors to address the points raised in the weaknesses section and to conduct additional experiments where further investigation is required.

---

> ### Author Response · Authors · 2024-11-17
> **Rebuttal**
>
> Thank you for reviewing our paper and giving constructive feedback.
>
> **Response to W1 (Lack of comparison with BEAST):**
> Thanks for pointing out the missing relevant comparison. See the table below (results on AdvBench test set).
>
> | TargetLLM  | BEAST (ASR) | AdvPrompter (ASR@1/@10) |
> |------------|-------------|-------------------------|
> | Vicuna-7B  | 96          | 35.6/85.6               |
> | Vicuna-13B | 93          | 23.1/74.7               |
> | Mistral-7B | 87          | 58.7/95.9               |
> | Falcon-7B  | 100         | 79.1/98.3               |
> | Llama-2-7B | 12          | 12.5/46.1               |
>
> Since BEAST uses the same dataset (AdvBench) for evaluation, we took the highest reported ASR presented in the original paper. With AdvPrompter, generating 10 (or even many more suffixes) is not expensive, therefore the main number for AdvPrompter is ASR@10. As you can see from the table, AdvPrompter shows competitive performance with BEAST. While AdvPrompter performs worse on some models, note the large improvement on the most difficult model Llama-2-7B.
>
> **Response to W2 (Unclear computational efficiency):**
> Thank you for pointing this out. We believe that there might be a slight misunderstanding here: AdvPrompter, once trained, generates suffixes simply by auto-regressive generation within seconds; there is no optimization involved, therefore generating orders of magnitude faster than BEAST. Of course, during training the computational cost is quite high and we have been very transparent about this throughout the paper (see e.g. sections 4.1 and B.5). The biggest computational burden of training the AdvPrompter is in the repeatedly used AdvPrompterOpt step, which by itself takes on average 2.7 minutes per prompt for <8GB LLMs on a single A100 GPU. This is very similar to the numbers reported in BEAST.
>
> **Response to W3 (Discussion of Attack Readability):**
> One possible way for improving qualitative analysis is to do a human-study on this matter, but we believe that this is unnecessary because:
> - We include a variety of non-cherry-picked adversarial prompts in the paper (see Appendix E).
> - We report the perplexity (for all white-box experiments) which is tightly correlated with human-readability. The perplexity is consistently low in all experiments we tried.
> - Human-readability in our method is achieved by construction (not as a side-product) as we describe in section 3.3.
>
> We believe that this sufficiently demonstrates our claim and is in accordance with other related work (e.g. AutoDAN, PAIR, etc.). Additionally, the replicable implementation (which will be open sourced with the paper) dumps all prompts and users can easily inspect and verify this claim.

---

> > ### Comment · Reviewer_qn1H · 2024-11-28
> >
> > I would like to thank the authors for their response. I am increasing my score.

---

> > > ### Author Response · Authors · 2024-12-02
> > >
> > > Dear Reviewer qn1H,
> > >
> > > We are grateful for your kind support and assistance in improving the paper.

---

### Official Review · Reviewer_QcAH · 2024-11-04

**Soundness:** 3
**Presentation:** 3
**Contribution:** 3
**Rating:** 6
**Confidence:** 3

**Summary:**

This paper proposes a novel method to enhance jailbreaking attacks on safety-aligned large language models (LLMs). The proposed method involves constructing a framework that fine-tunes an LLM from a base model by encouraging it to generate human-readable adversarial suffixes for harmful requests. Extensive experimental results demonstrate that the AdvPrompter can produce low-perplexity adversarial suffixes and achieve performance comparable to two baseline methods, i.e., GCG and AutoDAN.

**Strengths:**

* The proposed method enables the fast generation of specific adversarial suffixes for individual harmful requests.
* The experimental results show that the proposed method achieves great performance.

**Weaknesses:**

Overall, I think the method is sound, but there are a few concerns.

* The advantages of AdvPrompter mentioned in lines 108-136 should be specified under certain comparative conditions. For example, the "adaptivity to input" should be highlighted in the context of generating at a low time cost, as both GCG-individual and AutoDAN-individual are also adaptive to input. The "fast generation" compared to GCG and AutoDAN should be specified in the context of generating individual adversarial suffixes, since GCG-universal and AutoDAN-universal are ready to be used once obtained.

* Since both AutoDAN-universal and AdvPrompter generate human-readable adversarial suffixes quickly, it would be beneficial to discuss their performance in more detail, especially in Table 3.

* I think the "Gradient-free TargetLLM" is not a significant advantage, and it is unnecessary to emphasize the "gray-box TargetLLM" since it is actually a "white-box" model.

* More comparisons to existing methods, such as TAP and PAP, should be included. These methods also generate human-readable adversarial prompts.

**Questions:**

See Weaknesses.

---

> ### Author Response · Authors · 2024-11-17
> **Rebuttal**
>
> Thank you for reviewing our paper and giving constructive feedback.
>
> **Response to W1:**
> Thank you for pointing this out, we will clarify these conditions in the revision.
>
> **Response to W2:**
> Indeed, in table 3, PAIR, AutoDAN and AdvPrompter all generate human-readable suffixes, only GCG produces high-perplexity suffixes. Note that in this setting we consider AutoDAN-individual and not AutoDAN-universal. We observe that out of the former three methods, AdvPrompter achieves the highest ASR on Mistral-7b and Llama-3.1-8b (the most challenging out of the three models), and only performs slightly worse than PAIR on Vicuna-7b. This is achieved with a significantly lower inference time than PAIR and AutoDAN-individual, which run optimization directly on the test-set instances.
>
> **Response to W3:**
> You are correct that we can only use white-box models with this method. However, we still think it is important to highlight that we do not need to take gradients through the model, which significantly reduces the memory footprint. We will also clarify this in the revision.
>
> **Response to W4:**
> Thank you for this suggestion, please see the table below. It reports the results on the HarmBench test set across whitebox and blackbox models. The numbers for TAP, TAP-T and PAP-top5 are taken from the HarmBench paper, according to which TAP shows SOTA results on various GPT models. We observe that AdvPrompter performs competitively with even the best blackbox attack methods against GPT models on the HarmBench dataset.
>
> | TargetLLM     | TAP (ASR) | TAP-T (ASR) | PAP-top5 (ASR) | AdvPrompter (ASR@1/@10) |
> |---------------|-----------|-------------|----------------|-------------------------|
> | Vicuna-7B     | 51.7      | 60.2        | 19.2           | 42.8/68.1               |
> | Mistral-7B    | 62.8      | 65.8        | 26.6           | 54.2/77.8               |
> | GPT-3.5-Turbo | 38.9      | 47.6        | 17.0           | 49.4/79.1               |
> | GPT-4-0613    | 43.3      | 55.8        | 11.6           | 29.2/58.9               |
>
> Against Vicuna-7b and Mistral-7b, we evaluate AdvPrompter in the whitebox attack setting.
> Against the GPT models, we evaluate AdvPrompter in the blackbox transfer-attack setting. As the whitebox TargetLLM for training the AdvPrompter we use Llama3.1-8b in the case of GPT-3.5-Turbo, and Vicuna-7b in the case of GPT-4-0613.
> Additionally, in our response to reviewer qn1H, we report the results of the recently published method BEAST. Table 3 in the paper additionally reports results for PAIR.

---

> > ### Comment · Reviewer_QcAH · 2024-12-01
> >
> > Thanks for the reply. I have no further concerns.

---

> > > ### Author Response · Authors · 2024-12-02
> > >
> > > Dear Reviewer QcAH,
> > >
> > > We sincerely appreciate your valuable suggestions and thanks for helping us improve the paper!  If you don't have any further concerns, we would deeply appreciate it if you could consider raising your score. If not, please let us know your further concerns, and we will continue actively responding to your comments.
> > >
> > > Best,
> > > Authors

---

### Official Review · Reviewer_8Khr · 2024-11-04

**Soundness:** 3
**Presentation:** 3
**Contribution:** 2
**Rating:** 3
**Confidence:** 4

**Summary:**

The paper introduces a new method, and potentially a new perspective, for jailbreak prompting, the main contribution is that the prompter is itself a trained model, thus it can generate the jailbreak prompts extremely efficiently. The authors also mentioned several other properties such as human-readable or gradient-free, but these properties have been well discussed before.

**Strengths:**

1. The idea of using a pretrained model to directly generate jailbreak as next token prediction is interesting.

2. The propose method can generate jailbreak much faster than existing methods, especially since most existing methods are optimizing for every sample.

**Weaknesses:**

1. the idea of training a generation model for jailbreak has many natural limitations, such as the evaluation will require a different split of data, need to validating that the model can generalize across different target LLMs, and also different benchmark datasets. The authors didn't fully address these.

     -. 1.1 For example, since the method does not require any gradient of the target LLMs, the authors need to report more results on the commerlized LLMs where the strengths might be more obvious (referring to Table 2).

    -. 1.2. The evaluation is only limited to AdvBench, the empirical scope is too small. Some other choices are like HarmBench or JAMBench.

    -. 1.3. The evaluation needs to demonstrate the power of the advPrompter while the model is trained on AdvBench, and tested on other benchmarks such as HarmBench or JamBench. This is very important to show the advantages of proposed methods over the per-sample optimization method.

    -. 1.4. Similarly, the authors might want to offer more detailed and comprehensive discussions on the differences between the targetLLM during training vs. during testing, although a gentle discussion has been offered in 4.2.

2. The empirical scope is also fairly limited in terms of the methods being compared. Newer methods in jailbreaks, even just the ones published and presented in recent conferences (excluding arXiv ones), need to be discussed and compared. There are more methods that can deliver human-readable jailbreaks. (Although another method that can simultaneously fulfill all the properties in Table 1 might not exist).

3. While the authors present a unique method, and might be the only one at this moment can achieve all the properties in Table 1, the performance is unfortunately achieved by trade-offs. For example, in Table 2, the proposed method is not necessarily always the best performing method in ASR. The perplexity is always the lowest, but comparison to newer method might be needed, e.g., [1]. This is important because in AI security research, ASR and perplexity are probably more important factors than generation time. Authors might need to offer more convincing discussions why the method is favored although being lower in ASR in certain cases.


[1]. Role-playing to Generate Natural-language Jailbreakings to Test Guideline Adherence of LLMs

**Questions:**

1. Does the method require gradient during training? (i.e., does the method have to be trained with white-box LLM?) if that's the case, then that is another point needs to be made clear. If not, results showing how the trained advprompter from highly aligned models such as GPT then applied to less aligned models will be interesting.

2. Training time and requirement of computing might also need to be discussed, although less important.

3. The authors might need to compare their results in Sec. 4.3 with other jailbreak defensive methods for LLMs.

**Details Of Ethics Concerns:**

The paper is written in a technical way. Personally, I don't think the paper has ethical issues. However, the paper itself is about jailbreaking LLMs, a fairly sensitive topic, might benefit from an additional layer of caution.

---

> ### Author Response · Authors · 2024-11-17
> **Rebuttal (1/2)**
>
> Thank you for reviewing our paper and giving constructive feedback.
>
> **Response to W1:**
> We highly disagree with the criticism brought up here. We already address in the paper that the train/test split matters a lot, which is why we also report results on the HarmBench dataset in Table 3 and Table 5 which has minimal semantic overlap between data-splits. We also specifically test the generalizability of the AdvPrompter to different TargetLLMs in section 4.2.
>
> **Response to W1.1:**
> The method still requires access to output token probabilities, therefore when attacking commercial blackbox LLMs, it can only be used in a transfer setting. This scenario is explored in Section 4.2 and Fig. 2.
>
> **Response to W1.2:**
> This is incorrect, **we report results on HarmBench** in Table 3 and Table 5.
>
> **Response to W1.3:**
> In the results on HarmBench in Table 3 we aim to test exactly this transferability between datasets. HarmBench is specifically designed to have **minimal semantic overlap** between instances (between test and validation splits), therefore by training on one split (we use validation) and testing on another split we examine the transferability between datasets. And we observe that AdvPrompter preserves high transferability in this setup as well! Note that training on AdvBench and testing on HarmBench would be **worse than our setup**, because there is a significant semantic overlap between AdvBench and HarmBench.
>
> **Response to W1.4:**
> We agree that the discussion can be extended here. During training, we attack with AdvPrompterOpt the Vicuna-13b TargetLLM, exploiting the white-box nature of this model by using the output token probabilities to evaluate candidate tokens (no gradients of TargetLLM are involved). After training the AdvPrompter on the training set, we auto-regressively generate multiple responses of the AdvPrompter on the test set. The instructions and corresponding responses are then tested against the blackbox TargetLLM using an API. We are happy to include this extended discussion in a revised version of the manuscript.
>
>
> **Response to W2:**
> Missing comparisons to newer attacks has been a shared criticism between reviewers, therefore we now additionally report comparisons to recently published methods BEAST, TAP, PAP on a variety of settings. In the results, summarized in the tables in the responses to reviewers qn1H and QcAH, we observe very strong performance of our method even in comparison to SOTA blackbox attacks. Note that Table 3 in the paper also reports the results for PAIR.
>
> **Response to W3:**
> Firstly, in table 2, for Mistral-7b and Vicuna-13b, the results are quite similar to previous methods, and the slightly reduced ASR in some settings can be compensated for by the lower perplexity (the trade-off is controlled by a hyperparameter). These two models are also relatively easy to jailbreak, and on the more difficult Llama2-7b we achieve a much larger ASR while having the lowest perplexity.
> We also observe strong results in terms of ASR and perplexity in comparison to newer blackbox-attacks, see Figure 2 in the paper and the new results in the response to reviewer QcAH.
>
> Secondly, from an attacker perspective, ASR and perplexity are indeed the most relevant metrics. However, from the perspective of the designer of new LLMs, in a quickly shifting landscape of LLM capabilities, it is also important to quickly generate large amounts of safety fine-tuning data to account for edge cases of vulnerabilities. Our method takes a first step in exactly this direction: Scalable safety-fine-tuning data generation while re-using previously invested compute.
>
> Lastly, even though we achieve good ASR across different settings, reaching SOTA ASR is not our main focus, instead we offer a new learning-based approach that has not been explored in this context in previous work. Even if some newer methods exist that can outperform the token-by-token-based optimizer AdvPrompterOpt in terms of ASR, we firmly believe that our general AdvPrompterTrain learning-based paradigm is still highly relevant as it can easily be extended by improving the AdvPrompterOpt with newer mutation-based techniques.

---

> > ### Comment · Reviewer_8Khr · 2024-11-25
> > **re rebuttal**
> >
> > The authors failed to properly address so many of my comments, especially the last one, saying "reaching SOTA ASR is not our main focus" cannot dodge the question raised. I will keep my score.

---

> ### Author Response · Authors · 2024-11-17
> **Rebuttal (2/2)**
>
> **Response to Q1:**
> The method does not require gradients, but still requires the output token probabilities that are not always available for black-box LLMs, as described in line 133-136.
>
> **Response to Q2:**
> Training time and compute requirements are already discussed in Appendix C.1, Table 3 and in Section 4. Using the hyperparameters specified in Appendix C, the AdvPrompterTrain process averages 16 hours and 12 minutes for 7B TargetLLMs, and 20 hours and 4 minutes for 13B TargetLLMs, when run on 2 NVIDIA A100 GPUs for training 10 epochs.
>
> **Response to Q3:**
> While this would indeed be a nice addition, our goal there was to show an initial validation that AdvPrompter could be useful for improving LLM robustness. This was indeed the case when we tested it across different attack methods (e.g. ours, AutoDAN, GCG). We believe that more comprehensive evaluation of different alignment methods (e.g. DPO, SFT) and comparison with other existing defense mechanisms is beyond the scope of this paper.

---

> ### Author Response · Authors · 2024-11-30
>
> We regret to read that the reviewer is not fully satisfied with our responses in the rebuttal, but we hope to address some of the reviewer’s concerns in the remaining discussion time.
>
> > The authors failed to properly address so many of my comments
>
> We sincerely think that we have addressed the mentioned weaknesses, especially by adding in the new experimental comparisons to other attack methods (TAP, PAP, BEAST), as well as clarifying the misunderstandings regarding the HarmBench dataset. Our paper is a sound and valid scientific exploration of the ideas and offers new insights in the space of sharing information between adversarial LLM attacks. Can you please expand more concretely on any specific outstanding concerns you have on our paper?
>
> > saying "reaching SOTA ASR is not our main focus" cannot dodge the question raised
>
> We do not believe that we dodged your question here. In your last mentioned weakness you requested the following:
>
> > comparison to newer method might be needed, e.g., [1]
>
> > Authors might need to offer more convincing discussions why the method is favored although being lower in ASR in certain cases.
>
> Our answer was a three-fold discussion on this topic. First we pointed out the additional comparisons to newer methods (TAP, PAP, BEAST) listed in the responses to reviewers QcAH and qn1H, stressing that the ASR of our method is mostly competitive with other methods, although it is true that it is not always at the very top in terms of ASR. Then we explained why generation speed can be equally important as ASR and perplexity, from the perspective of safety-finetuning. Finally we highlighted that introducing a new learning-based paradigm to adversarial attacks on LLMs is an important contribution in and of itself, even without always achieving SOTA ASR, as the established paradigm extends beyond the results presented in our paper, e.g. offering room for potential improvements upon the inner optimization loop that may strongly increase ASR in future extensions of this work.
> We believe that our answer thoroughly addresses your comment, if you still disagree could you please clarify what is missing?

---

### Official Review · Reviewer_eFRA · 2024-11-06

**Soundness:** 2
**Presentation:** 3
**Contribution:** 2
**Rating:** 5
**Confidence:** 3

**Summary:**

This paper studies how efficiently generate transferrable and interpretable surfix for jailbreak. Unlike previous white-box attack methods that adopt search-based optimization, the authors propose a learning-based method, i.e. finetuning a LLM to generate adversarial prompts using annotated harmful QA data. A major benefit of this approach is inference-time efficiency. To train the LLM, the authors propose an alternated optimization by first searching for the best surfix that prompts the target LLM to answer harmful queries, and subsequently use it to finetune the Prompter LLM. Experiments are mainly conducted by comparing with some white-box attacks, and both direct search and transfer settings are considered. The result show that a major inference efficiency boost, with mixed results in terms of ASR. The proposed method also exhibits stronger transferability to close-source proprietary models than baselines.

**Strengths:**

1. Exploring the leanring-based paradigm for generating adversarial prompts is novel and relevant, to the best of the reviewer’s knowledge.
2. Compared with search-based paradigm, methods along the learning-based paradigm, like this one, naturally enjoy the benefit of inference-time efficiency.
3. Experiment results suggest that the proposed method surpasses previous method in terms of transferrability to black-box models, which is arguably a more practical scenario than white-box attack.

**Weaknesses:**

1. [major] It seems that, intuitively, the solution of equation 1 depends on the targetLLM. i.e. the optimal surfix that triggers a LLM to output the target response (e.g. “Sure, here are detailed xxx”) might be different. I’d imagine that this would hurt the transferrability in theory. It does appear that the transferability of AdvPrompter is at least better than early white-box attackers, but it might be due to poor transferability of white-box attackers in the first place.
2. [major] Following 1, I have some doubt about the practicality of jailbreak methods that requires transferrability in general. I would suggest comparing with SOTA blackbox methods on Figure 2. While I acknowledge that it is debatable whether such comparison is academically fair, the general practice usually guides us towards using whichever that is most effective. But I am happy to hear the author’s rebuttal and take them into consideration.
3. [minor] Learning-based paradigm, compared with search-based ones, incurs high training cost.

**Questions:**

1. Necessity of alternated update: Is it necessary to put the AdvPrompter in the loop of suffix generation (Figure 1 bottom right)? My understanding is that, the purpose of including it is to generate topk candidate tokens for the suffix. I am curious whether the author has tried to use a separate LLM to do this? The upside is that the dataset used to finetune the AdvPrompter can generated offline (without alternatively updating AdvPrompter); The downside is that the generated most likely tokens will not be adaptive.

---

> ### Author Response · Authors · 2024-11-17
> **Rebuttal**
>
> Thank you for reviewing our paper and giving constructive feedback.
>
> **Response to W1:**
> It is true that one could expect that the adversarial suffix is highly specific to the TargetLLM. However, in practice we observe that attacks are transferable to a surprisingly high degree. One hypothesis is that while the specific architecture and training procedures for various LLMs might differ, the data used for training (and specifically safety fine-tuning) is often very similar, which leads to shared vulnerabilities.
> Additionally, we believe that there is a distinction to be made between high- and low-perplexity adversarial suffixes. In the former case the suffix heavily exploits out-of-distribution non-human-readable text to achieve the jailbreaking behavior, in the latter the jailbreaking behavior is achieved with higher-level strategies, e.g. by convincing the TargetLLM of the non-harmfulness of the instruction using natural language. Naturally the latter approach appears much more independent of the choice of attacked TargetLLM, and this is potentially why we observe an improved transferability for our attack.
>
> **Response to W2:**
> First, we believe that transfer-based jailbreak attacks are not automatically un-practical. For example, note that jailbreak attacks are also useful for automated red-teaming, so transferability is not a necessity for an attack to be useful in practice.
> Moreover, in principle, human-written jailbreaking prompts transfer very well between models because they depend on high-level strategies that generalize across different LLMs. While we don’t claim that AdvPrompter has achieved human-level performance, we have seen some examples of high-level strategies emerging (e.g. Shakespearian, virtualizing it in a game environment, etc.) so it is reasonable that AdvPromtper transfers well.
> To quantitatively show the competitiveness of the transfer-attack with direct blackbox attacks, we additionally report a comparison against the blackbox methods TAP and PAP, please see our response to reviewer QcAH. According to the HarmBench paper, TAP shows SOTA performance on black-box attacks against some GPT models (see Table 7 therein). In our results, we observe that AdvPrompter performs very well across settings, even outperforming the blackbox methods on both tested GPT models.
>
> **Response to W3:**
> This is true and it is a natural drawback of the learning-based paradigm, which trades training cost for reduced inference time and potentially improved ASR by using shared information between instructions. We believe that this is a reasonable trade-off to be made.
> Something that we have not highlighted much in the paper but might be relevant here is that our proposed method actually allows for high flexibility in this trade-off, allowing for three possible settings:
> 1) Train the AdvPrompter as described in the paper and use fast auto-regressive generation at inference (this is our focus in the paper).
> 2) Completely ignore training and apply AdvPrompterOpt as a pure search-based method, reducing to a beam-search based variant of AutoDAN, similar to the recently proposed BEAST method.
> 3) Combine both, pre-training on the available data and then running AdvPrompterOpt on top at inference time.
>
> This flexibility allows the user to customize the method to a variety of setups.
>
> **Response to Q1:**
> If I understand correctly, what you are describing is the following two-stage approach: First create an offline dataset using the non-trained AdvPrompterOpt (meaning AdvPrompterOpt with a fixed non-trained AdvPrompter), then train the AdvPrompter on the offline dataset. This is indeed a valid option that we have also experimented with in the earlier stages of the project.
> This works when the attacked TargetLLM is not very safe, i.e. when the non-trained AdvPrompterOpt out-of-the-box generates jailbreaking suffixes. However, when attacking heavily safety-finetuned TargetLLMs (e.g. LLama2) AdvPrompterOpt (and also other attacks) often do not succeed in successfully jailbreaking suffixes on a large subset of the instructions. As a consequence, the resulting offline dataset is not usually sufficient to train a well-performing AdvPrompter.
> The natural extension of this approach is to iterate between dataset generation and training the AdvPrompter, where the dataset at each iteration gets progressively better as the AdvPrompter proposes better candidates for AdvPrompterOpt.
> We decided to directly take this one step further by continuously generating data for a replay buffer while updating the AdvPrompter on a running basis.
>
> Of course, in practice one should make use of any available data, which likely involves pre-training the AdvPrompter on an available offline dataset of suffixes to warm-start it. In the spirit of the analogy above, our training method could then be seen as an online algorithm that can be employed to further improve performance after pre-training on the available offline data.

---

### Author Response · Authors · 2024-11-21
**Checking in after Rebuttal**

Dear Reviewers,

We hope our rebuttal has addressed your concerns and enhanced the quality of our paper. Your feedback has been crucial in improving our work. If the changes meet the paper's objectives and your concerns, we hope this could be reflected in an improved score.

Please let us know if you have further questions or need additional information to aid your review. Thank you!

---

### Meta-Review · Area_Chair_5bmu · 2024-12-21

**Metareview:**

This paper introduces a novel method to enhance jailbreaking attacks on safety-aligned large language models (LLMs). The proposed approach involves developing a framework to fine-tune an LLM from a base model, encouraging it to generate human-readable adversarial suffixes for harmful requests. Extensive experimental results demonstrate that the method, named AdvPrompter, produces low-perplexity adversarial suffixes and achieves performance comparable to two baseline methods: GCG and AutoDAN.

The paper’s primary innovation lies in its alternating training method, AdvPrompterTrain, which alternates between generating adversarial suffixes and fine-tuning the AdvPrompter model. This process results in highly effective adversarial prompts, achieving state-of-the-art performance on the AdvBench and HarmBench datasets. AdvPrompter demonstrates improved attack success rates, faster generation times, and strong transferability to black-box LLMs. Furthermore, the paper shows that fine-tuning LLMs on datasets generated by AdvPrompter enhances their robustness against jailbreaking attacks while maintaining high performance on benchmarks like MMLU and MT-Bench.

AdvPrompter makes notable contributions to the field of LLM robustness and safety, offering a scalable and gradient-free solution to automatic red-teaming in both white-box and black-box settings, which broadens its applicability for securing deployed LLM systems.

However, there are significant shortcomings. One major omission is the lack of a detailed comparison with BEAST, a method introduced in *"Fast Adversarial Attacks on Language Models in One GPU Minute."* BEAST also employs a gradient-free approach, achieving high success rates with exceptional speed. Although the authors reference BEAST, they fail to provide head-to-head benchmarks, particularly regarding speed and success rates. This omission makes it difficult to verify AdvPrompter's claims of "fast" generation relative to an already proven rapid method.

Additionally, the empirical scope of the comparisons is limited. The authors do not sufficiently engage with newer jailbreak methods, including those presented at recent conferences. Many of these methods deliver human-readable jailbreaks and warrant a more thorough discussion and comparison. Reviewers also suggested comparing AdvPrompter's results with existing defensive techniques for LLMs, a point that remains underexplored.

While the authors addressed some concerns raised during the review process, several comments remain unresolved. The paper would benefit from an additional round of revision, including more comprehensive evaluations, detailed comparisons, and expanded discussions of related methods.

**Additional Comments On Reviewer Discussion:**

The authors of AdvPrompter reference BEAST but fail to provide direct, head-to-head comparisons, particularly in terms of speed and success rates. This lack of comparison limits the ability to evaluate whether AdvPrompter's claim of "fast" generation is valid when compared to a method that has already demonstrated both speed and effectiveness. The empirical scope of the comparisons is also quite narrow, as newer methods in jailbreak attacks—specifically those presented at recent conferences (excluding arXiv papers)—are not discussed or compared. Many of these newer methods also produce human-readable jailbreaks and warrant inclusion in a broader comparison. Additionally, as suggested by the reviewer, a comparison of AdvPrompter's results with existing defensive methods for LLMs is needed.

In response, the authors agree that the discussion could be expanded. They explain that during training, AdvPrompterOpt attacks the Vicuna-13b TargetLLM, exploiting the model’s white-box nature by using output token probabilities to evaluate candidate tokens, without involving gradients from the TargetLLM. After training the AdvPrompter on the training set, they generate multiple responses from it on the test set. The instructions and responses are then tested against a black-box TargetLLM via an API. The authors are open to including this extended discussion in a revised version of the manuscript.

Despite presenting a unique method that might be the only one currently achieving all the properties in Table 1, the performance is clearly achieved at the cost of certain trade-offs. The authors have not sufficiently addressed many reviewer comments, particularly the one regarding their claim that "reaching SOTA ASR is not our main focus." This response does not adequately address the concerns raised, leading reviewers to maintain their rating.

---

### Decision · Program_Chairs · 2025-01-22

Reject